# Structural basis for pharmacological modulation of the TRPC6 channel

Yonghong Bai[1†]*, Xinchao Yu[2†]*, Hao Chen[3], Daniel Horne[4‡], Ryan White[4], Xiaosu Wu[5], Paul Lee[6], Yan Gu[3], Sudipa Ghimire-Rijal[1], Daniel C-H Lin[5]*, Xin Huang[1]*

[1]Department of Molecular Engineering, Amgen Research, Amgen Inc, Cambridge, United States; [2]Department of Molecular Engineering, Amgen Research, Amgen Inc, South San Francisco, United States; [3]Department of Protein Technologies, Amgen Research, Amgen Inc, Cambridge, United States; [4]Department of Medicinal Chemistry, Amgen Research, Amgen Inc, Cambridge, United States; [5]Department of Cardiometabolic Disorders, Amgen Research, Amgen Inc, South San Francisco, United States; [6]Department of Discovery Technologies, Amgen Research, Amgen Inc, South San Francisco, United States

**Abstract** Transient receptor potential canonical (TRPC) proteins form nonselective cation channels that play physiological roles in a wide variety of cells. Despite growing evidence supporting the therapeutic potential of TRPC6 inhibition in treating pathological cardiac and renal conditions, mechanistic understanding of TRPC6 function and modulation remains obscure. Here we report cryo-EM structures of TRPC6 in both antagonist-bound and agonist-bound states. The structures reveal two novel recognition sites for the small-molecule modulators corroborated by mutagenesis data. The antagonist binds to a cytoplasm-facing pocket formed by S1-S4 and the TRP helix, whereas the agonist wedges at the subunit interface between S6 and the pore helix. Conformational changes upon ligand binding illuminate a mechanistic rationale for understanding TRPC6 modulation. Furthermore, structural and mutagenesis analyses suggest several disease-related mutations enhance channel activity by disrupting interfacial interactions. Our results provide principles of drug action that may facilitate future design of small molecules to ameliorate TRPC6-mediated diseases.

*For correspondence:
ybai80@gmail.com (YB);
xyu01@amgen.com (XY);
dclin@amgen.com (DCL);
hxin@amgen.com (XH)

[†]These authors contributed equally to this work

Present address: [‡]Department of Medicinal Chemistry, Skyhawk Therapeutics, Waltham, United States

## Introduction

The mammalian TRPC subfamily consists of seven transmembrane proteins (TRPC1-7) that have been proposed to form non-selective cation channels in various cell types (*Clapham et al., 2001*; *Venkatachalam and Montell, 2007*). TRPC6 and its most close homologs TRPC3 and TRPC7 are unique among the TRPCs in that they can be directly activated by second messenger diacylglycerol (DAG) (*Hofmann et al., 1999*), a product of phospholipase C action. TRPC6-mediated cation influx regulates physiological function of pulmonary endothelial cells (*Singh et al., 2007*), smooth muscle cells (*Dietrich et al., 2005*) and glomerular podocytes (*Reiser et al., 2005*), whereas TRPC6 hyperactivity has been implicated in maladaptive tissue and organ remodeling (*Reiser et al., 2005*; *Kuwahara et al., 2006*; *Onohara et al., 2006*; *Tian et al., 2010*; *Xie et al., 2012*; *Davis et al., 2012*). Notably, upregulation of TRPC6 in myocytes plays a role in cardiac hypertrophy (*Kuwahara et al., 2006*; *Onohara et al., 2006*; *Xie et al., 2012*; *Wu et al., 2010*), and gain-of-function mutations of TRPC6 contribute to hereditary focal segmental glomerulosclerosis (FSGS), a renal disorder characterized by podocyte injury and a potential cause of end stage renal disease (*Reiser et al., 2005*; *Heeringa et al., 2009*; *Ilatovskaya and Staruschenko, 2015*; *Mottl et al., 2013*; *Winn et al., 2005*).

Due to pathological roles of excessive TRPC6 activity, TRPC6 emerges as an important therapeutic target for pharmacological inhibition (*Bon and Beech, 2013*; *Lin et al., 2019*; *Seo et al., 2014*). However, development of potent and selective small-molecule antagonists of TRPC6 is hampered by limited understanding of the molecular mechanism of TRPC6 modulation. Recent cryo-electron microscopy (cryo-EM) structures of several TRPC channels elucidate their tetrameric assembly comprising a transmembrane domain and a cytoplasmic domain (*Azumaya et al., 2018*; *Duan et al., 2019*; *Duan et al., 2018*; *Fan et al., 2018*; *Sierra-Valdez et al., 2018*; *Tang et al., 2018*; *Vinayagam et al., 2018*). Despite identification of one antagonist-binding site in human TRPC6 (*Tang et al., 2018*) and multiple lipid-binding sites in human TRPC3 and mouse TRPC5 (*Duan et al., 2019*; *Fan et al., 2018*) molecular contacts with the antagonist are not well defined because of limited resolution, and the function of observed lipids remains to be determined (*Duan et al., 2019*; *Fan et al., 2018*; *Tang et al., 2018*). Furthermore, since there is no TRPC structure with a known agonist, it is still elusive how TRPC channels are activated by DAG or other agonists.

In this study, we present two high-resolution cryo-EM structures of antagonist- and agonist-bound human TRPC6 in lipidic nanodiscs. Our structures identify two novel modulation sites in the transmembrane domain and reveal the binding modes of the small molecule antagonist and agonist, which were corroborated by functional data. Conformational changes between these two structures allow us to gain further insights into the mechanism of TRPC6 function and modulation. Overall, these findings provide a rational basis for small molecule drug design for the treatment of TRPC6-mediated diseases.

## Results and discussion

### Characterization of N-terminally truncated TRPC6

An N-terminally truncated (Δ2–72) human TRPC6 was engineered because residues corresponding to 2–72 of TRPC6 are missing in TRPC3 despite high overall sequence similarity between TRPC3 and TRPC6. TRPC6 (Δ2–72) showed enhanced biochemical stability and could be activated by oleoyl-2-acetyl-sn-glycerol (OAG), a soluble analog of the native lipid agonist DAG (*Figure 1—figure supplement 1a*). We also evaluated the activity of TRPC6 (Δ2–72) in the presence of the antagonist AM-1473 and the agonist AM-0883 (*Figure 1—figure supplement 1b–d*). AM-1473, a small molecule antagonist with an IC50 of $0.22 \pm 0.05$ nM (n = 14) in our TRPC6 bioassay, is a structural analog of previously disclosed small molecule antagonist SAR-7334 (*Maier et al., 2015*). AM-0883, a novel TRPC6 agonist with an $EC_{50}$ of $45.5 \pm 10$ nM (n = 19) was identified from a small molecule in vitro high throughput screening campaign at Amgen. Similar to the WT protein, TRPC6 (Δ2–72) could be inhibited by AM-1473 (IC50 = $0.13 \pm 0.03$ nM, n = 5, *Figure 1—figure supplement 1c*) and activated by AM-0883 (EC50 = $90.2 \pm 13$ nM, n = 4, *Figure 1—figure supplement 1a–b*). These properties make TRPC6 (Δ2–72) suitable for further structural determination.

### Antagonist-bound structure of TRPC6

Detergent-solubilized TRPC6 (Δ2–72) was first purified to homogeneity in the presence of the antagonist AM-1473 (*Maier et al., 2015*), and then exchanged into lipidic nanodiscs composed of the membrane scaffold protein, MSP2N2, and soybean lipids. The structure of the antagonist-bound TRPC6 in nanodiscs was determined at a resolution of 3.1 Å using single-particle cryo-EM (*Figure 1*, *Table 1* and *Figure 1—figure supplements 2–3*). The overall architecture of the antagonist-bound TRPC6 is similar to previous cryo-EM structures of TRPC3 and TRPC6 (*Fan et al., 2018*; *Tang et al., 2018*). The voltage-sensor-like domain (S1-S4) and the pore domain (S5-S6) are arranged in a domain-swapped manner (*Figure 1*). S3 extends substantially into the extracellular space (*Figure 1a, c*). An elbow-like structural component is embedded in the lipid bilayer and makes hydrophobic contacts with the intracellular half of S1 (*Figure 1a,c*). The intracellular domain is assembled through interactions between the ankyrin repeat domain (ARD) at the N-terminus and the rib helix and coiled-coil at the C-terminus (*Figure 1a,c*).

### Antagonist-binding site

There are several non-protein densities in our cryo-EM map (*Figure 1*). Based on the shape and chemical environment, it is very likely that the density within the pocket of S1-S4 observed in the

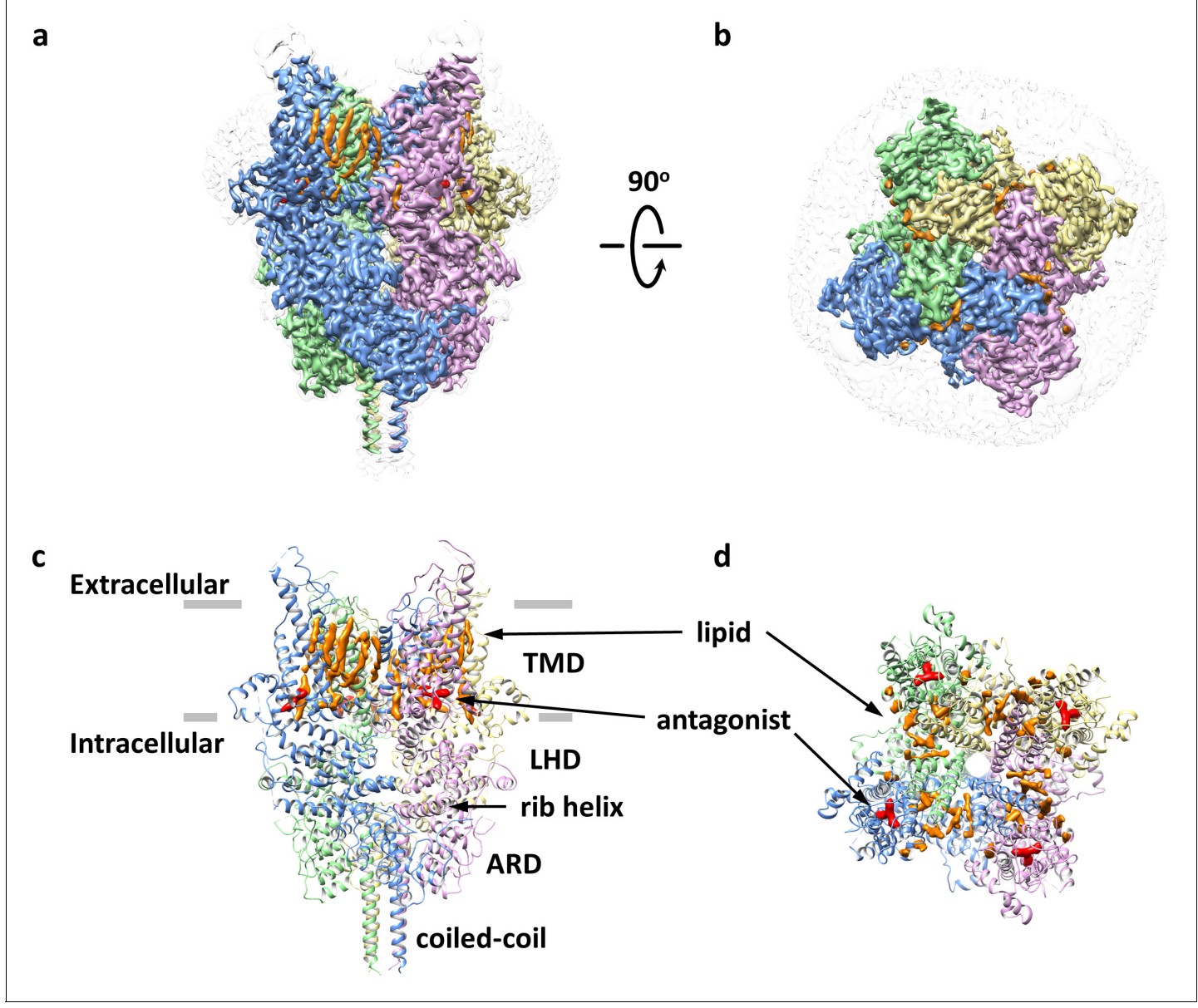

**Figure 1.** Overall architecture of the antagonist-bound TRPC6. (a, b) Cryo-EM map of the antagonist-bound TRPC6 viewed from parallel of the membrane (a) and the extracellular surface (b). The unsharpened reconstruction was shown in transparent gray. Lipids are colored in orange, and antagonists in red. (c, d) Structure model shown in corresponding orientations as in (a) and (b). Cryo-EM densities for lipids and antagonists are superimposed over the model and colored as in a and b.

The online version of this article includes the following figure supplement(s) for figure 1:

**Figure supplement 1.** Functional characterization of wild-type and (Δ2–72) TRPC6.

**Figure supplement 2.** Cryo-EM analysis of antagonist-bound TRPC6.

**Figure supplement 3.** Representative densities for the antagonist-bound TRPC6 reconstruction.

final postprocessed map and two corresponding half-maps belongs to the antagonist AM-1473 (*Figure 1c–d*, *Figure 2a–b*, and *Figure 1—figure supplement 2f–h*), which consists of an aminopiperidine, a benzonitrile, and an indane (*Figure 2a*). Furthermore, this pocket was empty in the structure of TRPC6 bound to another small molecule antagonist BTDM (*Tang et al., 2018*) or the TRPC3 structure (*Tang et al., 2018*; *Figure 2—figure supplement 1*), further supporting our assignment of the density to the antagonist. The antagonist engages in both hydrophilic and hydrophobic interactions with residues from S1-S4, the TRP helix and the membrane-reentrant loop following the TRP

**Table 1.** Cryo-EM data collection, refinement and validation statistics.

| | TRPC6-AM-1473 EMD-20954 PDB 6UZA | TRPC6-AM-0883 EMD-20953 PDB 6UZ8 |
|---|---|---|
| **Data Collection/processing** | | |
| Microscope | Titan Krios (FEI) | Titan Krios (FEI) |
| Voltage (kV) | 300 | 300 |
| Defocus range (mM) | −2.3 to −0.8 | −2.3 to −0.8 |
| Exposure length (s) | 6 | 6.2 |
| Electron exposure (e⁻/Å) | 50 | 50 |
| Number of frames | 30 | 31 |
| Pixel size (Å) | 0.832 | 1.248 |
| Initial particles images (no.) | 1,590,640 | 1,209,330 |
| Final particles images (no.) | 90,014 | 68,553 |
| Resolution (Å) | 3.08 | 2.84 |
| FSC threshold | 0.143 | 0.143 |
| Symmetry imposed | C4 | C4 |
| **Refinement** | | |
| Model composition | | |
| Protein residues | 2936 | 2936 |
| Ligands | 20 | 16 |
| R.m.s. deviations | | |
| Bond length (Å) | 0.005 | 0.005 |
| Bond angle (°) | 0.813 | 0.823 |
| Ramachandran plot | | |
| Favored (%) | 95.97 | 95.56 |
| Allowed (%) | 4.03 | 4.44 |
| Disallowed (%) | 0 | 0 |
| Validation | | |
| MolProbity score | 1.29 | 1.28 |
| Clashscore | 2.30 | 1.99 |
| Poor rotamers (%) | 0 | 0 |
| CaBLAM outliers (%) | 3.48 | 2.51 |
| EMRinger score | 3.55 | 3.29 |

helix (*Figure 2c*). The primary amine moiety off the piperidine ring forms hydrogen-bond interactions with Glu509 on S2, and Asp530 on S3. The benzonitrile group is involved in a cation-π interaction with Arg758 on the reentrant loop as well as aromatic-stacking interactions with His446 on S1 and Tyr753 on the TRP helix. The indene double ring makes van der Waal contacts with Tyr612 on S4. Overall, the three ring groups in the antagonist knit the intracellular ends of S1-S4, the TRP helix and the reentrant loop together. Interestingly, the antagonist is 36-fold selective for TRPC6 over its closely-related homolog TRPC3 (IC$_{50}$ of 8.0 ± 2.2 nM, n = 3, *Figure 1—figure supplement 1e*). Among the residues that interact with the antagonist, most are identical between TRPC3 and TRPC6, but Arg758 on TRPC6 is replaced by a lysine on TRPC3. We tested whether exchange this residue between TRPC6 and TRPC3 would have any effect on the antagonist potency and found that the antagonist is 5-fold less potent in TRPC6 R758K and 3.5-fold more potent in TRPC3 K689R (*Figure 2—figure supplement 2*). Therefore, we propose that the lower antagonist potency in TRPC3 could be partly due to the less optimal position for the lysine to form the cation-π interaction with the benzonitrile group.

To validate the antagonist-protein interactions observed in the structure, we carried out mutagenesis studies. Mutations that removed the negatively charged residues that interact with the primary amine, E509A and D530A, almost completely abolished the response of TRPC6 to the antagonist, consistent with the critical roles of the negatively charged side chains in ligand recognition and the positively charged primary amine in maintaining antagonist potency (*Figure 2d*). Interestingly, mutation R609A, which disrupts the salt-bridge interaction between Arg609 and Asp530, also lowered the binding affinity of compound by ~10 fold (*Figure 2d*). Mutation R758A, which disrupts the cation-π interaction, lowered the binding affinity by ~300 fold (*Figure 2d*). Mutations K442A and N527A lowered the binding-affinity by ~15 fold, suggesting that Lys442 and Gln527 might be involved in hydrophilic interactions with the nitrile and the primary amine, respectively (*Figure 2d*). Overall, these mutagenesis data are consistent with the atomic interactions visualized in the structure.

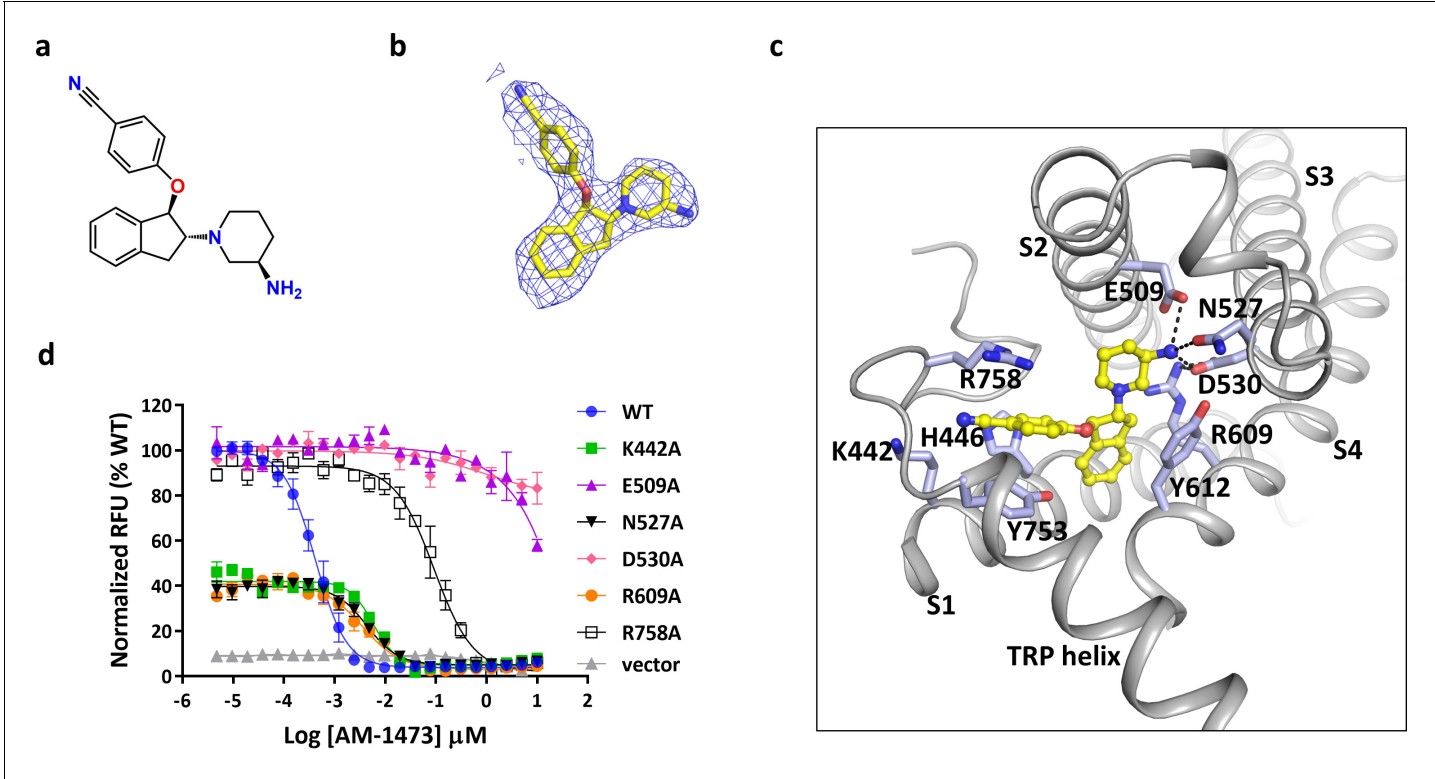

**Figure 2.** Binding of antagonist to TRPC6. (**a**) Chemical structure of the antagonist AM-1473. (**b**) Stick model of the antagonist model together with EM density depicted in blue mesh. (**c**) Close-up view of the antagonist-binding site. Residues within van der Waals distances are shown in light blue sticks. Hydrogen bonds are shown in black dashed lines. (**d**) Dose-response curves for TRPC6 inhibition of wild-type and mutant TRPC6 channels replacing residues that interact with the antagonist.

The online version of this article includes the following figure supplement(s) for figure 2:

**Figure supplement 1.** Density around the antagonist-binding site in the antagonist-bound TRPC6, agonist-bound TRPC6, BTDM-bound TRPC6 (PDB 5YX9), and TRPC3 (PDB 5ZBG) reconstructions.

**Figure supplement 2.** Dose-response curves for inhibition of TRPC6, TRPC6-R758K, TRPC3, and TRPC3-K689R by antagonist AM-1473 in the presence of OAG.

**Figure supplement 3.** Comparison of the modulation site located in the cytoplasm-facing pocket formed by S1-S4.

**Figure supplement 4.** Comparison of the lipid-binding site at the inner leaflet.

Interestingly, the S1-S4 pocket has been shown to play important roles in modulation of other TRP channels (*Figure 2—figure supplement 3*). In TRPV6, 2-aminoethoxydiphenyl borate was found to occupy this site and inhibit the channel activity (*Singh et al., 2018*; *Figure 2—figure supplement 3c–d*). On the other hand, in TRPM8, icilin, a synthetic cooling agonist, could bind to this site and increase the channel activity (*Yin et al., 2019*; *Figure 2—figure supplement 3e–f*). Since this pocket was empty in all other TRPC structures determined so far (*Duan et al., 2019*; *Duan et al., 2018*; *Fan et al., 2018*; *Tang et al., 2018*; *Vinayagam et al., 2018*), our structure of the antagonist-bound TRPC6 provides the first evidence that the S1-S4 pocket in TRPC channels could be a site for channel modulation. Notably, the crevice between the S2-S3 linker and the TRP helix connects the pocket to the cytoplasm and provides a possible access route for the antagonist.

## Lipid-binding sites

Besides the density for the antagonist, there are also several non-protein densities that most likely represent lipids copurified with TRPC6 or supplemented during purification. Similar to other TRP channels and voltage-gated ion channels, multiple outer leaflet lipids line the membrane-facing crevices between adjacent subunits, making contacts with helices S3-S6 as well as the pore helix (*Figure 1a,c*). In the inner leaflet, one lipid is buried in a pocket formed by S1, S4 and the pre-S1 elbow, and its size and shape fit well with that of a cholesterol hemisuccinate (CHS) molecule

(*Hughes et al., 2018*). Another inner leaflet lipid is wedged between S4 and the S4-S5 linker and can be interpreted as a phosphatidylcholine lipid (*Figure 2—figure supplement 3a*). While one acyl chain of the phospholipid is unresolved due to the interference of an outer leaflet lipid, the other acyl chain occupies the cleft formed by S4 and S5 from adjacent subunits (*Figure 2—figure supplement 3a*). Intriguingly, this cleft also contributes to the putative binding site of another small-molecule antagonist of TRPC6 (*Tang et al., 2018*; *Figure 2—figure supplement 3b*). Therefore, the intracellular cleft between S4 and S5 could be a site for channel modulation in TRPC6. Although it is not clear whether the inner leaflet phospholipid observed in our structure has an effect on channel activity, homologous lipid-binding sites in TRPV channels are critical for channel modulation. For example, the vanilloid-binding site located near S4 and S5 in TRPV1 accommodates a phosphatidylinositol lipid that could be displaced by vanilloid agonists or antagonists (*Gao et al., 2016*; *Figure 2—figure supplement 3d–e*). In TRPV5, econazole, a small molecule antagonist and antifungal, was found to occupy a similar site (*Hughes et al., 2018*; *Figure 2—figure supplement 3f*), whereas in TRPV6, native lipids that might be involved in channel activation were identified in this pocket (*McGoldrick et al., 2018*).

## Mapping of disease-related mutations

With the structure, we studied the location of disease-related mutations found in FSGS patients (*Mottl et al., 2013*). Intriguingly, many of these mutation sites are clustered at the region where the N-terminal ARD and the C-terminal rib helix and coiled-coil interact with each other (*Figure 3a–b*). Six mutations, including G109S, P112Q, N143S, Q889K, R895C, and E897K, are located at the interface between the ARD and the pole helix of the same subunit (*Figure 3b*, interface 1). Another mutation, M132T, is located at the interface between the ARD of one subunit and the rib helix of the adjacent subunit (*Figure 3b*, interface 2). Consistent with previous reports (*Heeringa et al., 2009*; *Mottl et al., 2013*; *Winn et al., 2005*), we found several mutations at interface 1, including those at positions 109, 112, 889, and 895, led to enhanced channel activity without affecting the expression level on the membrane (*Figure 3c*, *Figure 3—figure supplement 1a*). Since mutations at these positions destabilize the interactions at interface 1, the increased activity in mutant channels suggests that these interactions are important for the stability of the closed state of the channel.

To further explore this idea, we tested whether disrupting the interfacial interactions at interface 2 would have an effect on channel activity. M132T is a disease-causing mutation that also increases channel activity (*Figure 3d*). In our structure, Met132 forms hydrophobic interactions with Val867 and Leu868 on the rib helix. Substitution of either Val867 or Leu868 with a threonine increased maximum channel activity by ~50%, whereas mutating both residues simultaneously more than doubled maximum channel activity (*Figure 3d*). Since none of these mutations have any significant effect on the expression level on the membrane (*Figure 3—figure supplement 1b*), the increased channel activity of these mutant channels suggests that the interactions at interface 2 are also involved in stabilizing the closed state of TRPC6.

## Agonist-bound structure of TRPC6

To understand agonist binding and channel activation, we purified the double mutant TRPC6 (Δ2–72) V867T/L868T channel in complex with the small molecule agonist AM-0883, reconstituted it into nanodiscs and determined the cryo-EM structure at 3.1 Å (*Figure 4*, *Table 1* and *Figure 4—figure supplements 1–2*). AM-0883 induced about the same level of maximum activity as OAG, but was ~70 fold more potent than OAG (*Figure 4—figure supplement 1a–b*). We reasoned that the low potency of OAG might explain the lack of OAG density in the previous structure of TRPC3 copurified with OAG (*Tang et al., 2018*), and the higher potency of AM-0883 would allow us to obtain the agonist-bound structure of TRPC6. Furthermore, because TRPC6 V867T/L868T, like WT, responded to both AM-0883 and OAG but exhibited much higher channel activity than WT (*Figure 3d* and *Figure 1—figure supplement 1f*), the double mutant appeared to be an ideal variant for us to investigate the more activated state of TRPC6.

## Agonist-binding site

We found a clear non-protein density at each subunit interface that represents the agonist AM-0883, which consists of a chloro-indole, a piperidine and a benzodioxin, and occupies a groove between

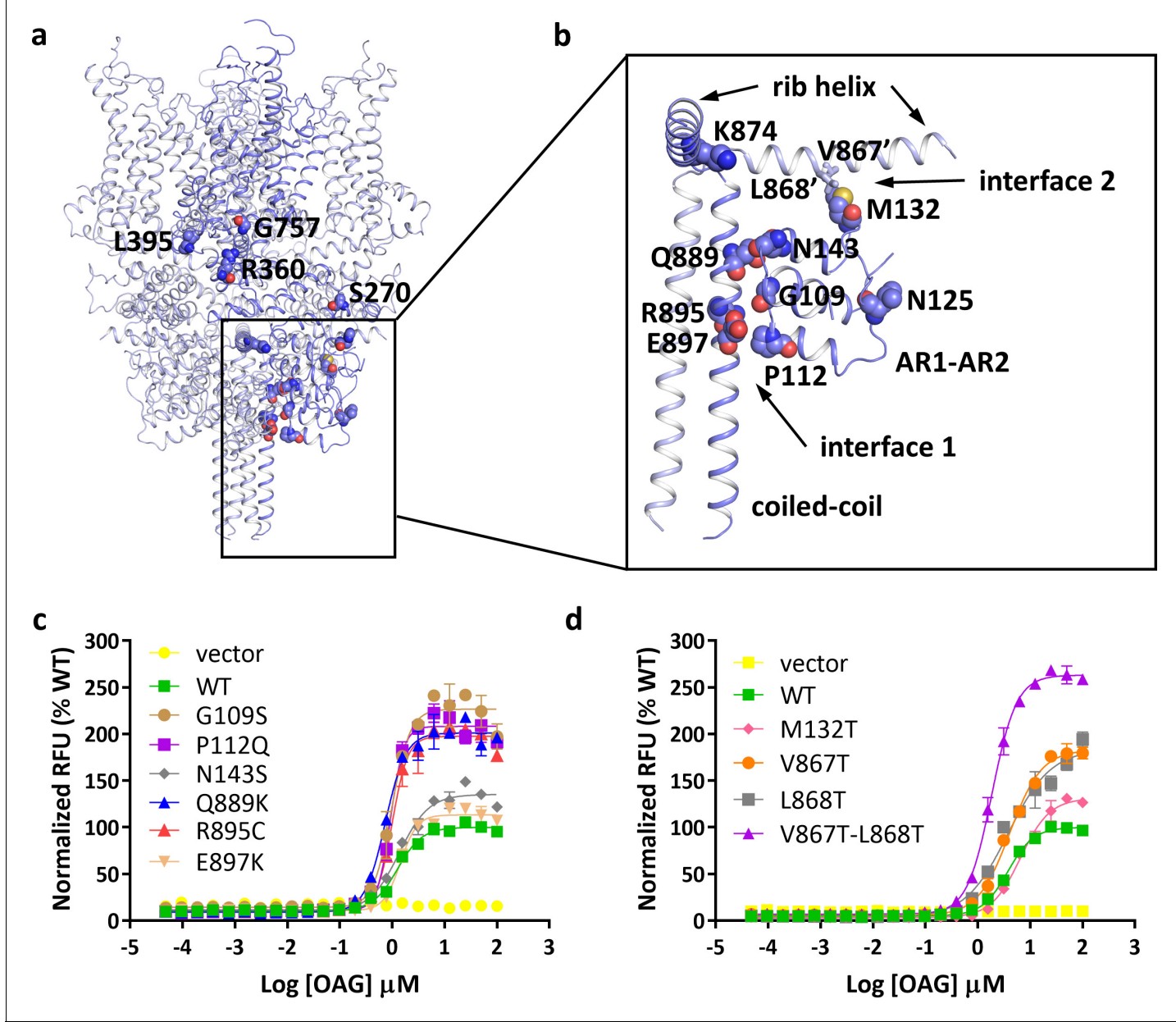

**Figure 3.** Location of FSGS-related mutations. (**a**) Overall structure with one subunit shown in blue and other three subunits in light blue. Residues whose mutations cause FSGS are shown as spheres on one subunit. (**b**) Close-up view of disease-related residues around the first two ankyrin repeats and C-terminal helices as boxed in (**a**). Side chains of V867 and L868, which form hydrophobic interactions with M132 in the adjacent subunit, are shown as sticks. (**c, d**) Dose-response curves for OAG activation of wild-type and mutant TRPC6 channels. Mutations located at interface 1 are shown in **c**, and mutations located at interface 2 are shown in **d**.

The online version of this article includes the following figure supplement(s) for figure 3:

**Figure supplement 1.** Surface expression level of TRPC6 wild type and mutants.

S6 of one subunit and the pore helix of the adjacent subunit (*Figure 4b–c* and *Figure 5a–b*). Similar densities also exist in the two half-maps generated during the final step of refinement but not in the BTDM-bound TRPC6 or the TRPC3 reconstructions (*Tang et al., 2018*), and the unique shape and size of these densities is distinct from what was observed near the same region in the maps of TRPC6-AM-1473 (*Figure 5—figure supplement 1*), indicating that this non-protein density belongs to the agonist. The agonist forms hydrophobic interactions with Phe675 and TRP680 on the pore helix and Tyr705, Val706 and Val710 on S6 (*Figure 5c*). There are also possible hydrophilic

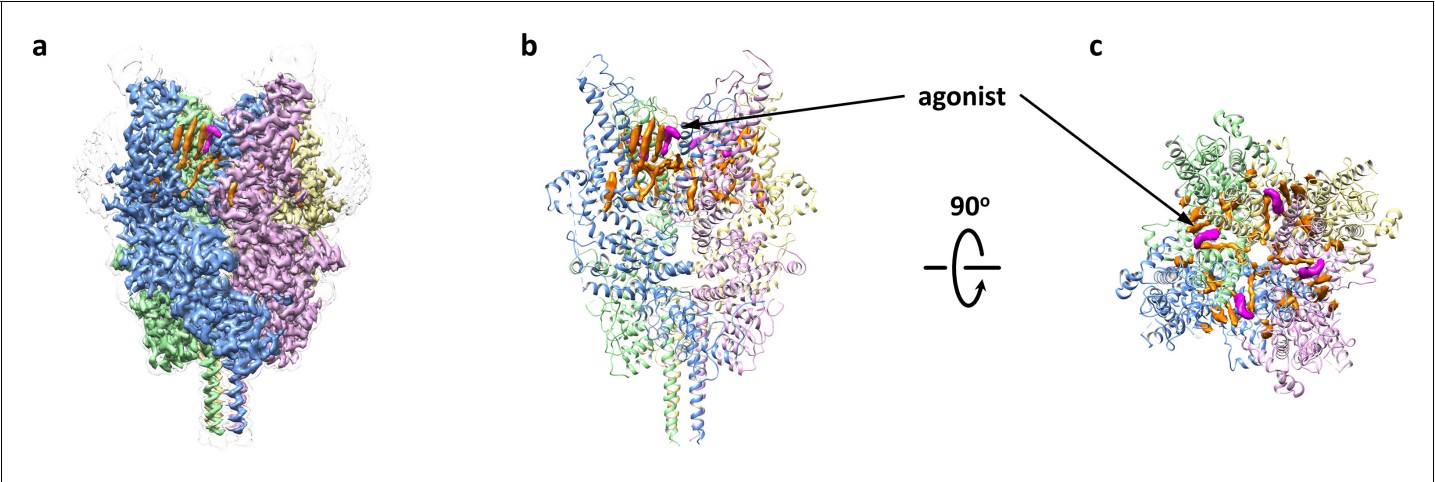

**Figure 4.** Overall architecture of the agonist-bound TRPC6. (a) Cryo-EM map of the agonist-bound TRPC6 viewed from parallel of the membrane. The unsharpened reconstruction was shown in transparent gray. Lipids are colored in orange, and agonists in magenta. (b, c) Structure model viewed from parallel of the membrane (b) and the extracellular domain (c). Cryo-EM densities for lipids and antagonists are superimposed over the model and colored as in a.

The online version of this article includes the following figure supplement(s) for figure 4:

**Figure supplement 1.** Cryo-EM analysis of agonist-bound TRPC6.

**Figure supplement 2.** Representative densities for the agonist-bound TRPC6 reconstruction.

interactions between the indole ring and Glu672 and Asn702 (*Figure 5c*). We tested whether mutations around this region would affect the potency of the agonist and found that mutations F675A, W680A, N702A, Y705A and V710A almost completely abolished the channel activity in the presence of the agonist AM-0883, whereas mutations E672A and V706A showed a 35-fold and 20-fold increase in agonist $EC_{50}$, respectively (*Figure 5d*). However, the expression level on the cell membrane of these mutants were similar to WT (*Figure 3—figure supplement 1c*), suggesting that the altered responses of the mutant channels to the agonist were most likely due to the disruption of the agonist-binding site.

Several residues around the agonist-binding site have been shown to affect channel activation by the native lipid agonist DAG. Gly640 on S6 of TRPC3, conserved among TRPCs and corresponding to Gly709 of TRPC6, has been proposed as a critical structural component for DAG recognition (*Lichtenegger et al., 2018*). Mutations of this glycine in TRPC3 and TRPC6 dramatically reduced DAG sensitivity (*Hofmann et al., 2002*; *Strübing et al., 2003*). On the pore helix, the conserved LFW motif (residues 678–680 in TRPC6) has been identified to be essential for channel activation (*Hofmann et al., 2002*; *Strübing et al., 2003*). Substitution of all three residues with alanine in TRPC5 and TRPC6 resulted in nonfunctional channels without altering plasma membrane expression (*Strübing et al., 2003*). Consistent with the idea that the agonist-binding site is also important for DAG recognition, mutations around the site, including E672A, F675A, W680A, N702A, Y705A and V706A, maintained normal membrane surface expression (*Figure 3—figure supplement 1c*) but completely eliminated the response to OAG even at a 100 µM concentration (*Figure 5e*). Furthermore, Glu672 and Val706 appeared to be more critical for the binding of DAG than AM-0883, because channels with mutation E672A or V706A could still be activated by the agonist AM-0883 at reduced potency but not by DAG (*Figure 5d*). The V710A mutation also reduced the maximal activation by OAG by >80% (*Figure 5e*). Importantly, among six TRPC proteins found in human, only the closely-related subgroup of TRPC3, TRPC6 and TRPC7 are directly activated by DAG (*Hofmann et al., 1999*). Indeed, whereas several residues around the putative DAG-recognition site are conserved across TRPCs, Glu672, Asn702 and Tyr705 are only conserved among TRPC3, TRPC6 and TRPC7 (*Figure 5—figure supplement 2*). Notably, the AM-0883-binding site faces membrane lipids and is occupied by one of the outer leaflet lipids in the antagonist-bound structure (*Figure 4—figure supplement 1a,c*), and therefore constitutes a likely location for DAG-binding at the outer

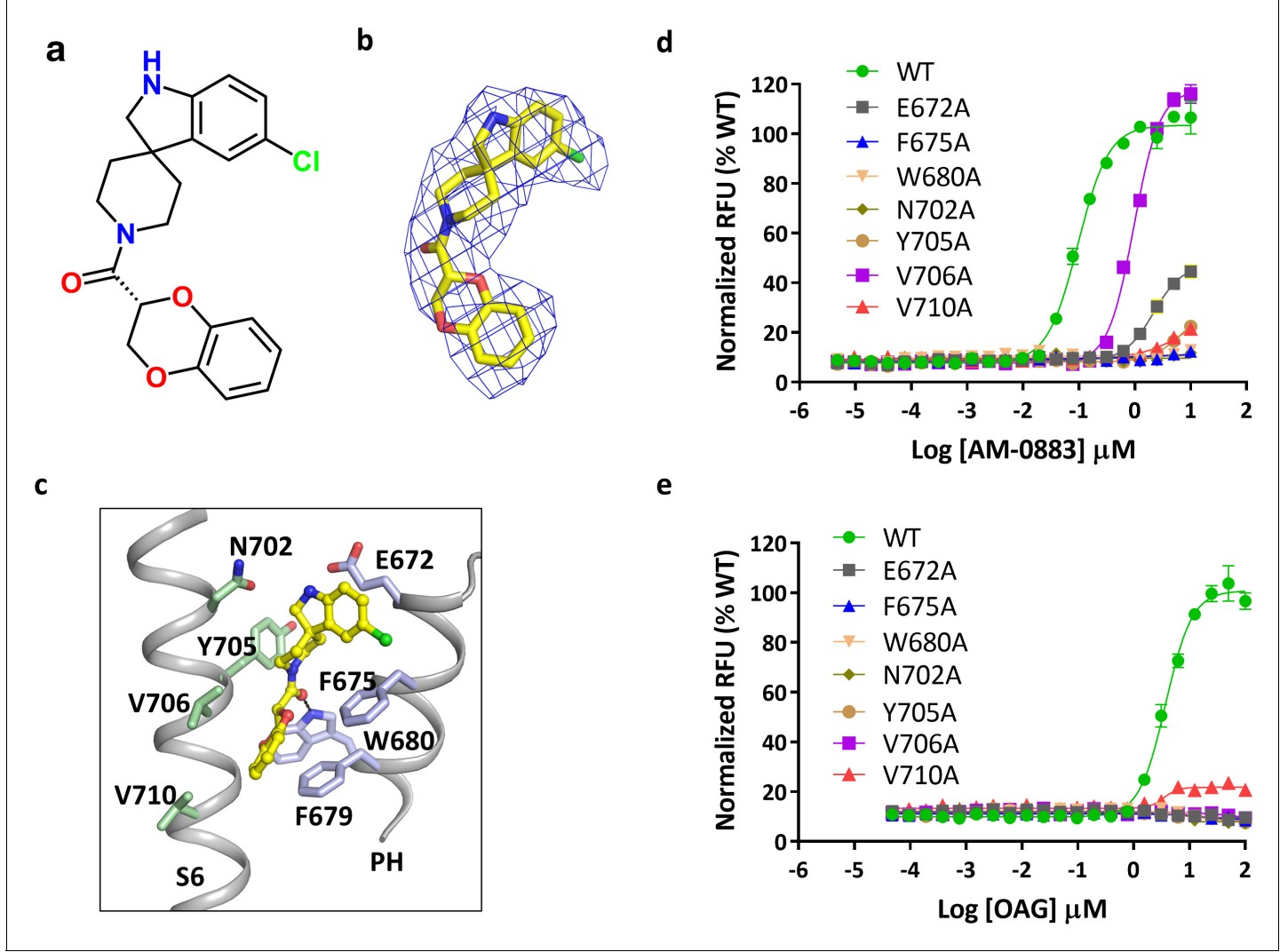

**Figure 5.** Binding of agonist to TRPC6. (**a**) Chemical structure of the agonist AM-0883. AM-0883 was synthesized as a racemate and then separated into enantiomers with arbitrarily assigned stereochemistry. The two enantiomers showed a 50-fold difference in potency in our bioassay. Since both enantiomers could be fit into the cryo-EM density, furthur work is needed to differentiate them. Here only the R-configuration is described. (**b**) Stick model of the agonist together with EM density depicted in blue mesh. (**c**) Close-up view of the agonist-binding site. Residues within van der Waals distances are shown in light blue sticks for one subunit and green sticks for another subunit. Hydrogen bonds are shown in black dashed lines. (**d, e**) Dose-response curves for TRPC6 activation by agonists AM-0883 (**d**) and OAG (**e**) of wild-type and mutant TRPC6 channels replacing residues that interact with the agonist.

The online version of this article includes the following figure supplement(s) for figure 5:

**Figure supplement 1.** Density around the agonist-binding site in the agonist-bound TRPC6, antagonist-bound TRPC6, BTDM-bound TRPC6 (PDB 5YX9), and TRPC3 (PDB 5ZBG) reconstructions.

**Figure supplement 2.** Sequence alignment of human TRPCs for the two helices that contribute to the binding-site of agonist.

**Figure supplement 3.** Comparison of the modulation site located at the subunit interface in the TMD.

leaflet. The overall findings strongly suggest that the agonist-binding site identified here may overlap with the DAG-recognition site.

Analogous sites near the subunit interface between S6 and the pore helix have been previously identified to accommodate a small molecule agonist in TRPML1 (*Schmiege et al., 2017*), and a small molecule antagonist in Ca$_v$Ab (*Tang et al., 2016*; *Figure 5—figure supplement 3*), corroborating the important role of this site in channel modulation. In addition, ligand binding at this site may also have an impact on lipid binding at the other side of the membrane, because in our agonist-bound structure, displacement of the outer leaflet lipid by the agonist appears to eliminate the steric

hindrance against one of the acyl chains in the inner leaflet lipid, thereby allowing both acyl chains to be visualized (*Figure 2—figure supplement 4a,c*).

## Ion permeation pathway

Compared to the WT protein, the double mutant V867T/L868T exhibited a 2-fold increase in maximum activity in the presence of the agonist AM-0883 (*Figure 1—figure supplement 1f*). However, similar to the antagonist-bound structure of TRPC6 (Δ2–72), the agonist-bound structure of TRPC6 (Δ2–72) V867T/L868T also possesses a closed ion channel pore (*Figure 6*). Therefore, the mutations and the agonist were not sufficient to shift the gating equilibrium towards a stable open state. This is not surprising because the low open probability and the short open time (<1 ms) of TRPC6 (*Hofmann et al., 1999*) suggest that the closed state is much more energetically stable compared to the open state. The ion permeation pathway in our structures is sealed off by hydrophobic side chains of Leu723, Ile727 and Phe731 near the intracellular end of S6 (*Figure 6a–b*), which are at equivalent positions in TRPC3 as shown in the human TRPC3 structure (*Fan et al., 2018*), but shifted by one amino acid in the previous human TRPC6 structure (*Tang et al., 2018*). On the other hand, the location of the selectivity filter in our structures is the same as or equivalent to what was observed before (*Fan et al., 2018*; *Tang et al., 2018*). In both our structures, main chain carbonyls of Phe683 and Gly684 located after the pore helix define the selectivity filter and coordinate a putative cation bound to the selectivity filter (*Figure 6a–b*).

## Ligand-induced structural changes

Although the ion channel pore is closed and the intracellular domain is nearly identical in both our structures (*Figure 7—figure supplement 1*), there are significant conformational changes in the transmembrane domain associated with ligand binding. In the ligand-free S1-S4 pocket of the agonist-bound state, the side chain of H446 adopts two alternative rotameric conformations, one of which occupies the position where the indene double ring of the antagonist would bind, and the side chain of R758 occupies the position where the benzonitrile group of the antagonist would bind (*Figure 7—figure supplement 2a*). Therefore, side chain movements of H446 and R758 are necessary to accommodate the antagonist. On the other hand, agonist-binding occurs without major structural changes in the side chains of the agonist-binding pocket (*Figure 7—figure supplement 2b*). Instead, the presence of the agonist seems to tilt the extracellular half of S6 away from the pore and pushes S6 toward the intracellular side (*Figure 7a*). Because S6 is tightly packed against S1-S4 of the adjacent subunit, the movement of S6 is accompanied by a concurrent tilting and downward movement of S1-S4 as well as the S4-S5 linker (*Figure 7a,c*). Among the transmembrane helices, S3

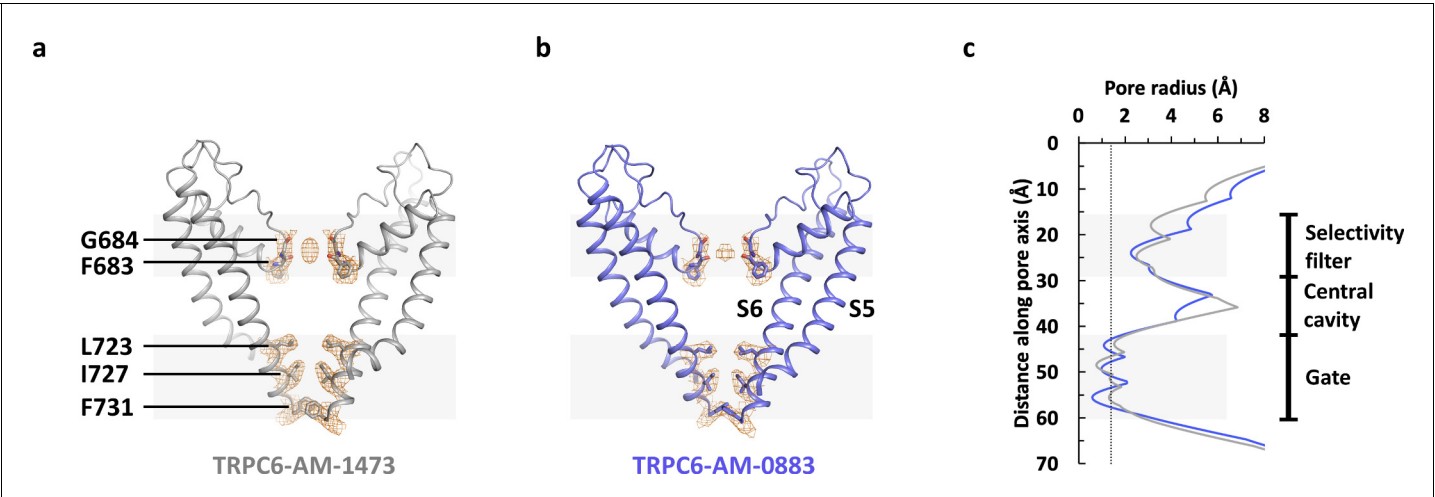

**Figure 6.** Ion channel pore. (a, b) Pore domain (S5-S6) of antagonist-bound (a) and agonist-bound (b) TRPC6 with front and rear subunits removed for clarity. Residues that form the selectivity filter and the intracellular gate are shown as sticks. Cryo-EM densities for these residues are shown as orange mesh. (c) Calculated pore radius along the central axis corresponding to a and b.

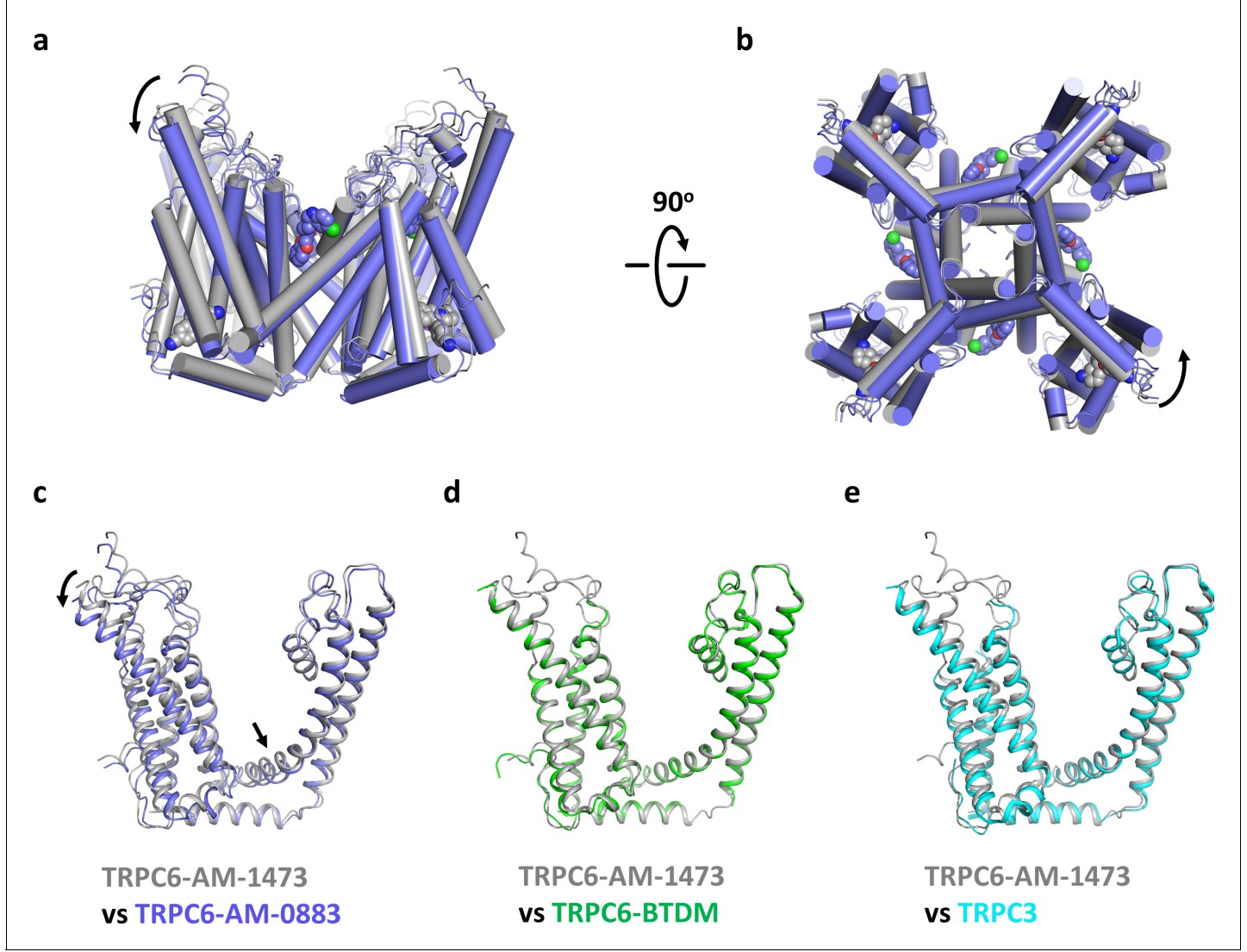

**Figure 7.** Conformational changes upon ligand binding. (a, b) Structural superposition of the transmembrane domain of antagonist-bound and agonist-bound TRPC6 viewed from parallel of the membrane (a) and intracellular side (b). (c-e) Structure of one subunit of the transmembrane domain in AM-1473-bound TRPC6 superposed with that of AM-0883-bound TRPC6 (c), BTDM-bound TRPC6 (d, PDB 5YX9), and TRPC3 (e, PDB 5ZBG). The structures are aligned with the pore helices. Black arrows indicate conformational changes from antagonist-bound state to agonist-bound state. Antagonists and agonists are shown as spheres.

The online version of this article includes the following figure supplement(s) for figure 7:

**Figure supplement 1.** Superposition of the intracellular domain between TRPC6-AM-1473 and TRPC6-AM-0883 (a), TRPC6-BTDM (b, PDB 5YX9) and TRPC3 (c, PDB 5ZBG).

**Figure supplement 2.** Structural comparison around the antagonist- and agonist- binding sites.

has the largest movement, with its extracellular end moving 4.4 Å, partly due to the remarkably long extension of S3 into the extracellular side. Viewing from the cytoplasmic side, agonist binding is accompanied by a counterclockwise rotation around the central ion pathway (*Figure 7b*), resulting in a movement of 2.4–3.3 Å at the intracellular end of S1-S4. While the structural changes from the antagonist-bound state to the agonist-bound state were not sufficient to stabilize an open channel pore, it is possible that agonist-bound channels may transition to the open conformation when further counterclockwise rotation of the S6 helical bundle fully unwinds the hydrophobic seal at the intracellular gate (*Video 1*).

To further understand how the antagonist inhibits TRPC6 activity, we also compared our structures with two previous TRPC structures prepared in a similar nanodisc environment (*Tang et al.,*

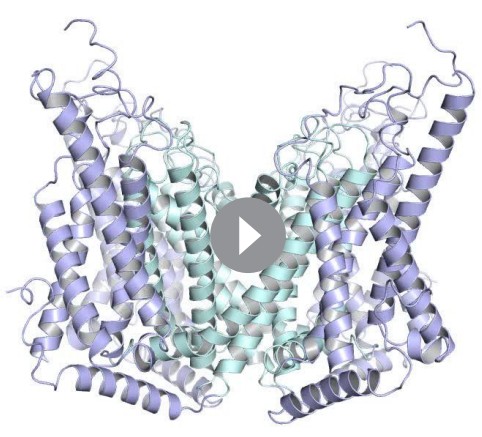

**Video 1.** Ligand-induced conformational changes. The video shows a morph of the TMD of TRPC6 from the antagonist-bound state to the agonist-bound state. S1-S4 and the TRP helix are colored in light blue, whereas S5-S6 are colored in light cyan. The video starts with the side view of the antagonist-bound TRPC6. The morph viewed at this orientation highlights the slight outward and downward movement of S1-S4. Next, the camera rotates to the intracellular view. The morph viewed at this orientation highlights the counterclockwise rotation of the S6 helices.
https://elifesciences.org/articles/53311#video1

*2018*). One of them is the structure of TRPC6 bound to another small molecule antagonist BTDM, and the other is the structure of TRPC3, which was prepared with OAG but no density of OAG was detected and the channel pore was closed. We found that the AM-1473-bound TRPC6 and the BTDM-bound TRPC6 assume a nearly identical conformation (*Figure 7d*, *Figure 7—figure supplement 1b*, Cα-RMSD 0.588), with S1-S4 and the S4-S5 linker in the slightly up conformation, distinct from the slightly down conformation in agonist-bound structure. The structure of TRPC3, which shares ~75% sequence identity with TRPC6, is also very similar to the AM-1473-bound and BTDM-bound TRPC6 structures (*Figure 7e* and *Figure 7—figure supplement 1c*, Cα-RMSD 0.710), suggesting that both antagonists may allosterically inhibit the opening of the channel pore by stabilizing the apo or resting state, as observed in the action of antagonists on P2X$_3$ receptor channels and TRPM4 channels (*Huang et al., 2020*; *Mansoor et al., 2016*). It would be interesting to test whether AM-1473 inhibits the channel by stabilizing the binding of the lipid near the S4-S5 linker, as this lipid and BTDM share part of their binding pockets and both seem to hinder the movement of the S4-S5 linker (*Figure 2—figure supplement 4a–b*). Future studies on a fully open TRPC6 channel would help test this idea.

## Conclusions

Our structures reveal novel binding modes of small molecule modulators of TRPC6 that allow us to delineate intriguing similarities and differences between the ligand-binding pockets in TRPC6 and those found in TRPV and TRPM channels. First, the antagonist-binding site at the cytoplasm-facing pocket in the S1-S4 domain has been found to recognize natural cooling agonists and various synthetic compounds in TRPM8 (*Yin et al., 2019*; *Diver et al., 2019*) as well as a synthetic antagonist in TRPV6 (*Singh et al., 2018*). Therefore, activity at the pore domain (S5-S6) of these channels could be allosterically modulated at the S1-S4 domain, most likely through the S4-S5 linker and the TRP helix. Second, the antagonist-binding site described here is distinct from the previously identified antagonist-binding site of TRPC6 (*Tang et al., 2018*), which is located at the intracellular membrane-facing cavity between S3-S4 and S5-S6 from adjacent subunits. Instead, this cavity is occupied by a phospholipid in both our structures and has been found to accommodate natural vanilloid agonists as well as competitive lipid or synthetic antagonists in TRPV1 (*Gao et al., 2016*), suggesting that occupancy by lipids or ligands at this site could also modulate channel activity. Finally, the agonist-binding site in TRPC6 is located at the extracellular membrane-facing cavity formed by S6 and the pore helix from adjacent subunits. We propose that this location is also where the native lipid agonist, DAG, acts on TRPC6. Therefore, TRPC6 is activated through a unique mechanism compared to TRPV and TRPM channels. Overall, our structures of TRPC6 bound to two different classes of modulators reveal hot spots in TRPC6 that future drugs could target and underline the emerging potential of cryo-EM in structural pharmacology.

## Materials and methods

### TRPC6 cloning and expression

An N-terminally truncated human TRPC6 (residue 73–931) and its variant V867T/L868T were each cloned into a pORBMam vector with an N-terminal Strep tag. The recombinant baculoviruses were generated in Sf9 cells following a conventional protocol. P2 virus was used to infect HEK293 cells lacking N-acetylglucosaminyltransferase I (GnTI⁻) at $3.5 \times 10^6$ cells/ml. After 12 hr of culture at 37℃, 10 mM Sodium butyrate was added to the suspension and the temperature was lowered to 30℃. Cells were harvested 48 hr post infection.

### TRPC6 purification and nanodisc reconstitution

All purification steps were done at 4℃ and in the presence of 1 μM antagonist or agonist. The cell pellet was resuspended in buffer A (150 mM NaCl, 20 mM Tris pH 8.0) supplemented with 0.5% (v/v) protease inhibitor cocktail. Cells were disrupted in an Microfluidizer and membrane fractions were isolated with two-step centrifugations. Membranes were first homogenized in buffer A and then solubilized with 1% (w/v) lauryl maltose neopentyl glycol (LMNG) and 0.1% (w/v) cholesteryl hemisuccinate (CHS) at 4℃ for 2 hr. Insoluble material was removed by centrifugation at 40,000 g for 1 hr and the supernatant was mixed with Strep Tactin resin at 4℃ for overnight. The resin was collected on a gravity column, washed with buffer A plus 0.06% (w/v) digitonin. The bound protein was eluted with 5 mM Desthiobiotin, concentrated and further purified by size exclusion chromatography using a Superose 6 Increase column equilibrated in buffer A plus 0.06% digitonin and 2 mM Tris(2-carboxyethyl)phosphine (TCEP). For nanodisc reconstitution, peak fractions were collected, concentrated to 1 mg/ml, and mixed with MSP2N2 and soybean lipid extract at a molar ratio of 1:3:225 for 1 hr. To remove detergents, two batches of fresh Bio-Beads SM2 were added at a concentration of 20 mg/ml with 4 hr in between. After overnight incubation, the sample was filtered and loaded onto a Superose 6 Increase column equilibrated in buffer A plus 2 mM TCEP. Peak fractions were pooled and concentrated to 1.5 mg/ml in the presence of 30 μM antagonist or agonist.

### Cryo-EM sample preparation and data collection

Grid preparation was performed at 100% humidity and 10℃ using a Mark IV Vitrobot (FEI). 3.5 μl of TRPC6 in nanodiscs was applied onto a glow-discharged Quantifoil R1.2/1.3 300-mesh copper holey carbon grid. Grids were blotted for 6 s at a force setting of 1 before being plunged into liquid ethane. Images were recorded on a 300 kV Titan Krios (FEI) microscope with a K2 summit detector (Gatan). Serial EM (*Mastronarde and Held, 2017*) was used for automated image acquisition with a binned pixel size of 0.832 Å. For the antagonist-bound TRPC6 dataset, 10,009 movies were collected, and each movie was dose-fractionated to 30 frames with a total exposure time of 6 s and a total dose of ~50 electrons/Å (*Venkatachalam and Montell, 2007*). For the agonist-bound TRPC6 dataset, 9517 movies were collected, and each movie was dose-fractionated to 31 frames with a total exposure time of 6.2 s and a total dose of ~50 electrons/Å (*Venkatachalam and Montell, 2007*).

### Cryo-EM data processing

Beam-induced motion was corrected in MotionCor2 (*Zheng et al., 2017*). Contrast transfer function (CTF) parameters were estimated non-doseweighted micrographs on using CTFFIND4 (*Rohou and Grigorieff, 2015*). All other data processing steps were performed using Relion-3 (*Zivanov et al., 2018*). Initially, 20,000 particles were autopicked using Laplacian-of-Gaussian method. After 2D classification, 8 class averages were selected for reference-based autopicking on the full dataset. The extracted particles were binned to a pixel size of 4.16 Å and subjected to two rounds of 2D classification. For the antagonist-bound TRPC6 dataset, 547,081 good particles were sorted out and used for subsequent 3D classification and refinement. The initial reference map was generated ab initio and lowpass filtered to 40 Å. One good 3D class out of three, containing 90,014 particles, were re-extracted to a pixel size of 1.248 Å. 3D refinement with C4 symmetry yielded a 3.26 Å map. After CTF-refinement and Bayesian-polishing, the final resolution was improved to 3.08 Å. For the agonist-bound TRPC6 dataset, 341,431 particles belonging to good 2D class averages were selected.

3D classification further sorted out 68,553 particles. After CTF-refinement and Bayesian-polishing, the final 3D refinement yielded a 2.84 Å map.

## Model building

The antagonist-bound TRPC6 model was built in Coot (*Emsley et al., 2010*) using the TRPC6 cryo-EM structure (*Tang et al., 2018*) (PDB 5Y × 9) as a guide. The model was subjected to real space refinement against sharpened map in Phenix (*Afonine et al., 2012*) with secondary structure restraints. The refined model of antagonist-bound TRPC6 was used as a reference to build the agonist-bound TRPC6 model. Local resolution was estimated using ResMap (*Kucukelbir et al., 2014*). Validation of geometries was performed in MolProbity (*Chen et al., 2010*). All the structure figures were generated in Chimera (*Pettersen et al., 2004*), Pymol (The PyMOL Molecular Graphics System) and HOLE (*Smart et al., 1996*).

## FLIPR assay

TRPC6 $Ca^{2+}$ channel activity was measured using a FLIPR (fluorescence imaging plate reader) Tetra system from Molecular Devices and the BD PBX Calcium Assay Kit (Becton Dickinson #640177). HEK293T cells were maintained in DMEM high glucose +10% FBS + 1X NEAA (Invitrogen #11965) and were transiently transfected with TPRC6 WT or variant expression plasmids. Site directed mutagenesis to create TRPC6 variants was carried out by Genewiz (South Plainfield, NJ) and variants were verified by DNA sequencing. Expression plasmids were prepared for transfection using Lipofectamine 3000 (Invitrogen) and added to cells. 15,000 cells/well were plated in a 384-well black poly-D-lysine coated plate (Corning #356663). 24 hr post transfection, cells were loaded with calcium sensitive fluorescent dye utilizing the BD PBX Calcium Assay kit following the manufacturer's protocol and incubated for 2 hr in the dark at room temperature. Compound plates were prepared in assay buffer containing 10 mM HEPES pH = 7.2 @25˚C, 4 mM $MgCl_2$, 120 mM NaCl, 5 mM KCl, 0.1% BSA, 2 mM $CaCl_2$. Compound addition to cells was automated on the FLIPR Tetra and fluorescent imaging was captured following the manufacturer's protocol (Molecular Devices). Data were analyzed using GraphPad Prism 7 software.

## Acknowledgements

We thank C Xu and K Song at the cryo-EM core facility at UMass Medical school for assistance in data collection. We also thank J Hu, H Zhao, Q Shi, and S Mukund for assistance in protein expression and purification. Lastly, we are grateful to L Miranda and P Tagari for leadership and support throughout the project.

## Additional information

### Competing interests

Yonghong Bai: At the time of the study YB was affiliated with Amgen Research, Amgen Inc and has no financial interests to declare. The author has no other competing interests to declare. Xinchao Yu: XY is affiliated with Amgen Research, Amgen Inc and has no financial interests to declare. The author has no other competing interests to declare. Hao Chen: At the time of the study HC was affiliated with Amgen Research, Amgen Inc and has no financial interests to declare. The author has no other competing interests to declare. Daniel Horne: At the time of the study DH was affiliated with Amgen Research, Amgen Inc and has no financial interests to declare. The author has no other competing interests to declare. Ryan White: At the time of the study RW was affiliated with Amgen Research, Amgen Inc and has no financial interests to declare. The author has no other competing interests to declare. Xiaosu Wu: XW is affiliated with Amgen Research, Amgen Inc and has no financial interests to declare. The author has no other competing interests to declare. Paul Lee: At the time of the study PL was affiliated with Amgen Research, Amgen Inc and has no financial interests to declare. The author has no other competing interests to declare. Yan Gu: At the time of the study YG was affiliated with Amgen Research, Amgen Inc and has no financial interests to declare. The author has no other competing interests to declare. Sudipa Ghimire-Rijal: SGR is affiliated with Amgen Research, Amgen Inc and has no financial interests to declare. The author has no other

competing interests to declare. Daniel C-H Lin: DCL is affiliated with Amgen Research, Amgen Inc and has no financial interests to declare. The author has no other competing interests to declare. Xin Huang: At the time of the study XH was affiliated with Amgen Research, Amgen Inc and has no financial interests to declare. The author has no other competing interests to declare.

## Funding
No external funding was received for this work.

## Author contributions
Yonghong Bai, Conceptualization, Data curation, Formal analysis, Supervision, Validation, Investigation, Visualization, Methodology, Project administration, Writing—original draft, Writing—review and editing; Xinchao Yu, Hao Chen, Daniel Horne, Ryan White, Daniel C-H Lin, Xin Huang, Conceptualization, Data curation, Formal analysis, Validation, Investigation, Methodology, Writing—review and editing; Xiaosu Wu, Yan Gu, Sudipa Ghimire-Rijal, Data curation, Formal analysis, Validation; Paul Lee, Data curation, Formal analysis, Validation, Investigation, Methodology

## Author ORCIDs
Yonghong Bai (iD) https://orcid.org/0000-0002-4334-0916

## Decision letter and Author response
Decision letter https://doi.org/10.7554/eLife.53311.sa1
Author response https://doi.org/10.7554/eLife.53311.sa2

# Additional files

## Supplementary files
• Transparent reporting form

## Data availability
The low pass filtered and amplitude modified 3D cryo-EM density maps for TRPC6 in complex with antagonist AM-1473 (accession code: EMD-20954) and agonist AM-0883 (accession code: EMD-20953) have been deposited in the electron microscopy data bank. Atomic coordinates for TRPC6 in complex with antagonist AM-1473 (accession code: 6UZA) and agonist AM-0883 (accession code: 6UZ8) have been deposited in the protein data bank.

The following datasets were generated:

| Author(s) | Year | Dataset title | Dataset URL | Database and Identifier |
|---|---|---|---|---|
| Bai Y, Yu X, Huang X, Chen H | 2019 | Cryo-EM structure of human TRPC6 in complex with antagonist AM-1473 | http://www.rcsb.org/structure/6UZA | RCSB Protein Data Bank, 6UZA |
| Bai Y, Yu X, Huang X, Chen H | 2019 | Cryo-EM structure of human TRPC6 in complex with antagonist AM-1473 | http://www.ebi.ac.uk/pdbe/entry/emdb/EMD-20954 | Electron Microscopy Data Bank, EMD-20954 |
| Bai Y, Yu X, Huang X, Chen H | 2019 | Cryo-EM structure of human TRPC6 in complex with agonist AM-0833 | http://www.rcsb.org/structure/6UZ8 | RCSB Protein Data Bank, 6UZ8 |
| Bai Y, Yu X, Huang X, Chen H | 2019 | Cryo-EM structure of human TRPC6 in complex with agonist AM-0833 | http://www.ebi.ac.uk/pdbe/entry/emdb/EMD-20953 | Electron Microscopy Data Bank, EMD-20953 |

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
