## [Decision Letter]

**Acceptance summary:**

This study presents structures of TRPC6 channels in complex with an agonist and an antagonist developed by Amgen, solved by single-particle cryo-EM. The structures reveal two novel binding sites – one for the antagonist and the other for the agonist in the transmembrane domain. The authors also propose that the agonist binding site may overlap with the OAG binding site, the location of which has been a mystery for lipid-sensitive TRPC3/6/7 channels. This report is an important advance, because TRPC6-specific antagonists or agonist have been lacking, and knowledge of the binding sites could be further exploited for structure-based drug design.

**Decision letter after peer review:**

Thank you for submitting your article "Structural basis for pharmacological modulation of the TRPC6 channel" for consideration by *eLife*. Your article has been reviewed by three peer reviewers, including László Csanády as the Reviewing Editor and Reviewer #3, and the evaluation has been overseen by Richard Aldrich as the Senior Editor. The following individuals involved in review of your submission have agreed to reveal their identity: Vera Y Moiseenkova-Bell (Reviewer #2).

The reviewers have discussed the reviews with one another and the Reviewing Editor has drafted this decision to help you prepare a revised submission.

Several concerns were raised by the reviewers which will need to be addressed to solidify the conclusions.

Essential revisions:

1) It would be important to solve the structure of the TRPC6 channel in the apo state to allow the authors to confidently assign ligand densities to the synthetic compounds used in this study, and to propose mechanisms of actions by comparing the apo structure to the ligand-bound structures. The apo structure should be obtained in the nanodiscs under the same experimental conditions as they did for ligand-bound structures. The same lipids and nanodiscs should be used for the preparation of this channel.

2) Interpreting changes in maximal whole-cell fluorescence level as changes in maximal channel activity is not justified as long as changes in channel surface expression are not determined. Along those lines, increased fluorescence in certain mutants, as well as reduced efficacies of the synthetic agonist and OAG to stimulate channels mutated at/around the agonist binding site, could as well be explained by increased and reduced surface expression, respectively, of the mutant channels. Specifically:

2.1) Subsection “Mapping of disease-related mutations”: The authors claim that mutations at the interface between the N-terminal ARD and the C-terminal rib helix and coiled-coil result in "increased channel activity". However, "maximal activity" is defined here based on fluorescence imaging on whole cells. Such an approach cannot discern effects on channel expression levels from effects on channel activity (open probability). At the minimum, the authors should show that the WT and the mutant channels were expressed at comparable levels at the cell surface, to support their claim.

2.2) Subsection “Agonist-binding site”: Mutations F675A, W680A, N702A, and Y705A eliminate channel activation by both OAG and the synthetic agonist. The authors interpret these data to suggest that this region is important for agonist binding. Again, surface expression levels of these mutants should be shown to remain comparable to WT in order to support this claim.

3) The authors should clearly explain the rational for determining the structure of the TRPC6(2-72)T867T/L868T mutant and not of TRPC6(2-72) in the presence of the agonist. Did they try and did not see the ligand in the TRPC6(2-72) structure?

4) Figure 7 is confusing. The authors used two similar colors for the antagonist-bound and agonist-bound structures to compare the conformational changes induced by agonist and antagonist. The authors should revise the figure. With reference to comment 1), they should also provide the structural comparison with the apo structure. Also, are there any conformational changes in the intracellular domain?

---

## [Author Response]

Essential revisions:1) It would be important to solve the structure of the TRPC6 channel in the apo state to allow the authors to confidently assign ligand densities to the synthetic compounds used in this study, and to propose mechanisms of actions by comparing the apo structure to the ligand-bound structures. The apo structure should be obtained in the nanodiscs under the same experimental conditions as they did for ligand-bound structures. The same lipids and nanodiscs should be used for the preparation of this channel.

We appreciate and agree with the reviewers’ comment on the importance of the apo-TRPC6 structure determined in the same condition as used in the ligand-bound structures. In fact, we did try extensively to obtain such a structure but without any success. The main reason is that the apo-TRPC6 protein (wild type or N-terminally truncated) is very unstable and not suitable for cryo-EM analysis. While we could not surpass the technical challenges in obtaining the apo-TRPC6 structure at this point, we carried out additional structural analysis to address the concerns mentioned in the comment.

A) Assignment of ligand densities. First, in Figure 2—figure supplement 1, we compared the density around the antagonist-binding site between TRPC6-AM-1473 (3.1 Å) and TRPC6-AM-0883 (2.8 Å), TRPC6-BTDM (3.8 Å) or TRPC3 (4.3 Å). TRPC6-AM-1473 and TRPC6-AM-0883 protein samples were prepared with the same nanodisc condition in our lab, and TRPC6-BTDM and TRPC3 protein samples were prepared with very similar nanodisc condition from an external lab. The AM-1473 density only exists in the TRPC6-AM-1473 reconstruction. All the other reconstructions show an empty pocket around this region. Second, in Figure 5—figure supplement 1, we used the same above reconstructions to compare the density around the agonist-binding site. Again, we only found the AM-0883 density in the TRPC6-AM-0883 reconstruction. The density in the TRPC6-AM-1473 reconstruction looks quite different from that in the TRPC6-AM-0883 reconstruction and most likely belongs to phospholipids. There is no density for either AM-0883 or lipids in the TRPC6-BTDM and TRPC3 reconstructions, which have a relatively lower resolution. Third, in Figure 1—figure supplement 2F-H and Figure 4—figure supplement 1F-H, we included ligand densities from the postprocessed map and two corresponding half maps. The densities in all maps match very well with the unique size and shape of the small molecules. Overall, with the added structural analysis, we are confident that these densities belong to the small molecule modulators.

B) Understanding of the mechanism. We also added a structural comparison of our two structures and two other TRPC structures determined in a very similar nanodisc reconstitution. As discussed in our main text (Conclusion section), the comparison suggests that AM-1473 may help stabilize the apo/resting state. We believe the newly added structural analysis further expand the structural insights into the action of the small molecule modulators of TRPC6. We admit that at this point it is still challenging to fully understand how exactly these two molecules work partly due to the lack of a fully open TRPC6 structure. However, our structures lay a foundation for future research on a more activated state that would help better understand channel function and modulation.

2) Interpreting changes in maximal whole-cell fluorescence level as changes in maximal channel activity is not justified as long as changes in channel surface expression are not determined. Along those lines, increased fluorescence in certain mutants, as well as reduced efficacies of the synthetic agonist and OAG to stimulate channels mutated at/around the agonist binding site, could as well be explained by increased and reduced surface expression, respectively, of the mutant channels. Specifically:2.1) Subsection “Mapping of disease-related mutations”: The authors claim that mutations at the interface between the N-terminal ARD and the C-terminal rib helix and coiled-coil result in "increased channel activity". However, "maximal activity" is defined here based on fluorescence imaging on whole cells. Such an approach cannot discern effects on channel expression levels from effects on channel activity (open probability). At the minimum, the authors should show that the WT and the mutant channels were expressed at comparable levels at the cell surface, to support their claim.2.2) Subsection “Agonist-binding site”: Mutations F675A, W680A, N702A, and Y705A eliminate channel activation by both OAG and the synthetic agonist. The authors interpret these data to suggest that this region is important for agonist binding. Again, surface expression levels of these mutants should be shown to remain comparable to WT in order to support this claim.

We agree with the reviewers’ comment. We carried out new experiments testing the surface expression levels of WT and all mutant channels mentioned in comments 2.1 and 2.2. The results showed that WT and all mutant channels were expressed at similar levels on the membrane surface (Figure 3—figure supplement 1). These results suggest that the increased channel activity and lack of activity we observed with mutant channels are most likely due to perturbation of channel function instead of channel expression and further support our interpretation of the assay data.

3) The authors should clearly explain the rational for determining the structure of the TRPC6(2-72)T867T/L868T mutant and not of TRPC6(2-72) in the presence of the agonist. Did they try and did not see the ligand in the TRPC6(2-72) structure?

We appreciate the reviewers’ comment and have modified the text (subsection “Agonist-binding site”) accordingly. The main reason of working on the agonist-bound structure with the double mutant is that we found that the double mutant channel has much higher activity compared to the WT channel and therefore would be a better variant to investigate the more activated conformation. On hindsight, unfortunately, the double mutant did not shift the gating equilibrium enough for us to observe the fully open channel. We did not further determine the agonist-bound structure with TRPC6 D2-72 because based on the similar potency of the agonist (Figure 3D and Figure 1—figure supplement 1F), we don’t expect there would be any significant difference in the binding mode between these two channel variants.

4) Figure 7 is confusing. The authors used two similar colors for the antagonist-bound and agonist-bound structures to compare the conformational changes induced by agonist and antagonist. The authors should revise the figure. With reference to comment 1), they should also provide the structural comparison with the apo structure. Also, are there any conformational changes in the intracellular domain?

We appreciate the reviewers’ comment. We have changed the colors in this Figure as well as in Figure 6 so that there is more contrast among the structures and included the comparison with the apo structure of TRPC3 (ref. response to comment 1). We also included structural comparison of the intracellular domain (Figure 7—figure supplement 1), which did not show any significant conformational changes.